# Two Novel *er1* Alleles Conferring Powdery Mildew (*Erysiphe pisi*) Resistance Identified in a Worldwide Collection of Pea (*Pisum sativum* L.) Germplasms

**DOI:** 10.3390/ijms20205071

**Published:** 2019-10-12

**Authors:** Suli Sun, Dong Deng, Canxing Duan, Xuxiao Zong, Dongxu Xu, Yuhua He, Zhendong Zhu

**Affiliations:** 1National Key Facility for Crop Gene Resources and Genetic Improvement, Institute of Crop Sciences, Chinese Academy of Agricultural Sciences, Beijing 100081, China; sulisun@caas.cn (S.S.);; 2Zhangjiakou Academy of Agricultural Sciences, Zhangjiakou 075000, China; 3Yunnan Academy of Agricultural Sciences, Kunming 650205, China

**Keywords:** *Erysiphe pisi*, *er1*-8, *er1*-9, KASPar marker, pea

## Abstract

Powdery mildew caused by *Erysiphe pisi* DC. severely affects pea crops worldwide. The use of resistant cultivars containing the *er1* gene is the most effective way to control this disease. The objectives of this study were to reveal *er1* alleles contained in 55 *E. pisi*-resistant pea germplasms and to develop the functional markers of novel alleles. Sequences of 10 homologous *PsMLO1* cDNA clones from each germplasm accession were used to determine their *er1* alleles. The frame shift mutations and various alternative splicing patterns were observed during transcription of the *er1* gene. Two novel *er1* alleles, *er1*-8 and *er1*-9, were discovered in the germplasm accessions G0004839 and G0004400, respectively, and four known *er1* alleles were identified in 53 other accessions. One mutation in G0004839 was characterized by a 3-bp (GTG) deletion of the wild-type *PsMLO1* cDNA, resulting in a missing valine at position 447 of the PsMLO1 protein sequence. Another mutation in G0004400 was caused by a 1-bp (T) deletion of the wild-type *PsMLO1* cDNA sequence, resulting in a serine to leucine change of the PsMLO1 protein sequence. The *er1*-8 and *er1*-9 alleles were verified using resistance inheritance analysis and genetic mapping with respectively derived F_2_ and F_2:3_ populations. Finally, co-dominant functional markers specific to *er1*-8 and *er1*-9 were developed and validated in populations and pea germplasms. These results improve our understanding of *E. pisi* resistance in pea germplasms worldwide and provide powerful tools for marker-assisted selection in pea breeding.

## 1. Introduction

Pea (*Pisum sativum* L.) is a widely distributed legume crop, which frequently suffers from various stresses, including abiotic and biotic factors in the season of growth [1,2]. Powdery mildew, induced by *Erysiphe pisi* DC., severely reduces the yield and quality of pea crops worldwide [3,4,5]. Severe *E. pisi* infections of peas can lead to yield losses of up to 80% in regions which are suitable for disease development [5,6]. The use of resistant cultivars carrying the *E. pisi*-resistant gene *er1* has been considered to be the most effective and environmentally friendly way to prevent this disease to date [6,7].

Formerly, *E. pisi* infection was the only known cause of pea powdery mildew. However, since 2005, two other *Erysiphe* species, *Erysiphe trifolii* and *Erysiphe baeumleri*, have been reported to also infect peas and induce the same powdery mildew symptoms as *E. pisi* in some regions [8,9,10]. Previous studies of pea powdery mildew have primarily focused on breeding peas resistant to *E. pisi*. Their results have indicated that resistance to *E. pisi* is controlled by two single recessive genes (*er1* and *er2*) and one dominant gene (*Er3*) [11,12,13,14]. The *er1*, *er2*, and *Er3* genes have been mapped using linked markers [15,16,17,18,19,20,21,22,23,24,25,26,27]. The genes *er1* and *er2* map to pea linkage groups (LGs) VI and III, respectively [17,28]. *Er3,* which was isolated from wild pea (*Pisum fulvum*), was initially mapped on an uncertain pea LG, but it was more recently assigned to pea LG IV [29].

As *er1* confers high resistance or complete immunity to *E. pisi* in most pea germplasms, it is currently the most widely used gene in pea production [30]. In contrast, *er2* is only found in a few pea germplasms resistant to *E. pisi* [30]. Moreover, the efficacy of *E. pisi* resistance conferred by *er2* varies with leaf development stage and plant location [12,30,31,32]. *Er3* was known from wild pea (*P. fulvum*), and there have not been extensive studies conducted to date [13,33].

Gene *er1* confers stable, durable, and broadly effective resistance to *E. pisi*. This gene inhibits the incursion of *E. pisi* into pea epidermal cells [32]. Recent studies have shown that the *er1*-resistant phenotype is caused by loss-of-function mutations in the pea MLO (Mildew Resistance Locus O) homolog (*PsMLO1*). The MLO gene family has been identified in both dicots (e.g., *Arabidopsis thaliana* and tomato: *Solanum lycopersicum*) and monocots (e.g., barley: *Hordeum vulgare*) [14,34,35,36,37,38,39].

To date, nine *er1* alleles resistant to *E. pisi* have been identified in *E. pisi*-resistant pea germplasms: *er1*-1 (also known as *er1mut1*) [14,21,25,40,41], *er1*-2 [14,24,25], *er1*-3 [14], *er1*-4 [14], *er1*-5 [38], *er1*-6 [27], *er1*-7 [26], *er1*-10 (also known as *er1mut2*) [21,40,42], and *er1*-11 [42,43]. Each *er1* allele corresponds to a different *PsMLO1* mutation site and pattern. Among the nine *er1* alleles identified, only *er1*-1 and *er1*-2 are commonly applied in pea breeding programs [14,38]. Several studies have attempted to design functional markers of *er1* alleles to allow for the rapid selection of pea germplasms resistant to *E. pisi* [24,26,27,38,42,43,44].

The yield and quality of the Chinese pea crop are severely damaged by powdery mildew [2], with the disease affecting up to 100% of pea plants in some regions of China [4]. Several studies have focused on the identification of Chinese pea germplasms resistant to *E. pisi* [41,44,45,46,47,48,49]. In the Chinese pea cultivars X9002 and Xucai 1, *E. pisi* resistance is conferred by the *er1*-2 allele [24,25,47], while in some Chinese pea landraces from Yunnan Province, *E. pisi* resistance is conferred by the *er1*-6 allele [27,48]. *E. pisi* resistance in the Indian pea cultivar DDR11 is conferred by the *er1*-7 allele [26]. Thus, natural resistance to *E. pisi* conferred by the *er1* gene has been observed in pea germplasms worldwide, providing a rich source of genetic material that can be used to improve the *E. pisi* resistance of Chinese pea cultivars [41,46,48,50]. Allelic diversity of this locus in the cultivated pea has been well characterized; however, relatively few studies have investigated and characterized *E. pisi*-resistant pea germplasms in an international collection. Thus, this study aimed to identify and characterize the *E. pisi*-resistant alleles at the *er1* locus in a worldwide collection of pea germplasms resistant to *E. pisi*. Additionally, any novel *er1* alleles were genetically mapped, and functional markers specific to these novel *er1* alleles were developed to improve marker-assisted selection in pea breeding programs.

## 2. Results

### 2.1. Phenotypic Evaluation

Fifty-five *E. pisi-*immune or -resistant pea germplasm accessions from 13 countries were re-evaluated for their resistance to the *E. pisi* isolate EPYN. At 10 days post-inoculation, the *E. pisi* disease severity of all susceptible controls (Bawan 6 and Longwan 1) were rated as score 4. In contrast, the 55 *E. pisi-*resistant germplasm accessions appeared to be either immune (symptom-free; disease severity 0) or resistant (slight infection; disease severity 1–2) to *E. pisi* isolate EPYN. Of the 55 resistant germplasm accessions, 46 were classified as immune and nine as resistant to *E. pisi* (Table 1). To provide comprehensive information for the resistance of a worldwide collection of 86 pea germplasms to *E. pisi*, the phenotypes of 31 resistant pea germplasms carrying known *er1* alleles are also shown in Table 1.

### 2.2. PsMLO1 Sequence Analysis

The *PsMLO1* cDNA sequence of Bawan 6 and Longwan 1, the susceptible controls, was consistent with that of the wild-type *PsMLO1* cDNA (Table 1). Among the 55 resistant pea germplasms with previously unknown *er1* alleles, *er1*-1 was identified in seven germplasm accessions, *er1*-2 in 37, *er1*-6 in seven, and *er1*-7 in two (Table 1 and Table 2).

Two novel *er1* alleles were discovered in the two remaining germplasms: G0004389 (from Afghanistan) and G0004400 (from Australia). A novel mutation pattern was found in the G0004389 cDNA fragment homologous to *PsMLO1*: a 3-bp deletion (GTG) corresponding to positions 1339–1341 in exon 15 (the final exon) of the *PsMLO1* cDNA sequence. This deletion caused the loss of the amino acid valine at position 447 of the PsMLO1 protein sequence, probably resulting in a functional change (Figure 1A). This mutation differed from all known *er1* alleles, indicating that the *E. pisi* resistance of G0004389 was controlled by a novel allele of *er1*. This novel allele was designated *er1*-8, following the accepted nomenclature [14,26,27,42,44,51]. In pea germplasm G0004400, a 1-bp deletion (T) was identified in a previously unreported position homologous to position 928 in exon 10 of the *PsMLO1* cDNA sequence. This deletion caused a substitution of the amino acid serine with leucine at position 310 of the PsMLO1 protein sequence (Figure 1B). This change caused the early termination of protein translation, probably also resulting in a functional change of PsMLO1 (Figure 1B). Thus, *E. pisi* resistance in G0004400 was also controlled by a novel *er1* allele, herein designated *er1*-9.

Interestingly, frame shift mutations, where small fragments are deleted or inserted, were identified in the cloned sequences of several pea germplasms. The fragments homologous to the wild-type *PsMLO1* cDNA in seven pea germplasms (G0002602, G0006515, G0002883, G0004448, G0002848, G0003935, and G0005117) had 5-bp deletions (GTTAG) at positions 700–704 of wild-type *PsMLO1* cDNA, while three pea germplasm accessions (G0002883, G0002971, and L0368) had another 5-bp deletion (TAGGG) at positions 1235–1239 of the wild-type *PsMLO1* cDNA. In accession G0006514, there was a 4-bp deletion (GGAG) at positions 181–184 of the wild-type *PsMLO1* cDNA. In four pea accessions (G0002847, G0004434, G0003974, and Texuan 11) and two pea accessions (G0002235 and G0002848), there were a 16-bp deletion (CTCATCTTCCTCCAGG) at positions 776–791 and a 16-bp insertion (AATTTTTCTGTTTCAG) at position 1171 of the wild-type *PsMLO1* cDNA, respectively. In germplasm accession Jia 2, there was a 7-bp insertion (TAATAAG) at position 921 of the wild-type *PsMLO1* cDNA. It was probable that these indels resulted from aberrant splicing events during transcription. Each frame shift mutation was observed in only one or two of ten cloned *PsMLO1* cDNA sequences per germplasm accession.

Various alternative splicing patterns, including intron retention and exon skipping, were also observed in multiple PsMLO1 sequences cloned from the 55 resistant pea germplasm accessions. The eight introns retained were 1, 2, 4, 6, 7, 9, 12, and 13, and the three exons skipped were 4, 10, and 11 of the wild-type PsMLO1. Each intron retention and exon skipping event were discovered in only one or two of ten cloned PsMLO1 cDNA sequences.

### 2.3. Genetic Analysis and Mapping of er1-8 and er1-9

As expected, the two resistant pea parents, G0004389 and G0004400, were immune to *E. pisi* infection (disease severity 0), while the two susceptible parents (Bawan 6 and WSU 28) were heavily infected (disease severity 4) (Figure 2). The segregation patterns of *E. pisi* resistance in the F_1_, F_2_, and F_2:3_ populations derived from the crosses WSU 28 × G0004389 and Bawan 6 × G0004400 are presented in Appendix A.

Six F_1_ plants produced from the cross WSU 28 × G0004389 were susceptible to *E. pisi* (Appendix A). One of the six plants generated 120 F_2_ and F_2:3_ offspring through self-pollination. Of these 120 F_2_ plants, 30 were resistant (R) to *E. pisi,* and 90 were susceptible (S) to *E. pisi-*. This indicates that the segregation ratio (resistance:susceptibility) in the F_2_ population was exactly 1:3 (χ^2^ = 0.01; *P* = 0.92), indicating recessive heredity of a single gene. Moreover, a segregation ratio of 30 (homozygous resistant): 63 (segregating): 27 (homozygous susceptible) in the F_2:3_ population fitted well with the genetic model of 1:2:1 ratio (χ^2^ = 0.48, *P* = 0.79) (Appendix A), confirming that the *E. pisi* resistance in G0004389 was controlled by a single recessive gene.

The cross of Bawan 6 × G0004400 generated five F_1_ plants which showed *E. pisi*-susceptibility (Appendix A). One of five F_1_ plants generated 119 F_2_ offspring. 32 of 119 were resistant, and 87 of 119 were susceptible to *E. pisi*. The segregation ratio in the F_2_ population of resistance to susceptibility fitted a genetic model ratio of 1:3 (χ^2^ = 0.14; *P* = 0.71), also indicating recessive heredity of a single gene. Moreover, a segregation ratio of 32 (homozygous resistant): 64 (segregating): 23 (homozygous susceptible) in the F_2:3_ population (119 families) fitted well with the genetic model of 1:2:1 ratio (χ^2^ = 2.51; *P* = 0.29), indicating that *E. pisi* resistance in G0004400 was also controlled by a single recessive gene (Appendix A).

Of the 20 markers tested, five (c5DNAmet, AD160, AA200, AA224, and PSMPSA5) were polymorphic between parents WSU 28 and G0004389, and seven (AC74, AD160, PSMPSAD51, ScOPD10-650, ScOPX04-880, ScOPE16-1600, and AD59) were polymorphic between Bawan 6 and G0004400, indicating that these markers were likely linked to the *E. pisi* resistance gene. Thus, the five and the seven parental polymorphic markers were used to confirm the genotypes of each F_2_ plant derived from WSU 28 × G0004389 and Bawan 6 × G0004400, respectively. This genetic linkage analysis suggested that three markers (c5DNAmet, AA200, and AA224) and six markers (AD160, PSMPSAD51, ScOPD10-650, ScOPX04-880, ScOPE16-1600, and AD59) were linked to the resistance gene *er1* in G0004389 and G0004400, respectively (Figure 3). Our results also indicated that the resistance genes in both germplasm accessions were located in the *er1* region. In G0004389, the linkage map indicated that the markers (c5DNAmet and AA200) were mapped on both sides of the target gene with 9.6 cM and 3.5 cM genetic distances, respectively (Figure 3A). In G0004400, two other markers (PSMPSAD51 and ScOPX04-880) were located on both sides of the target gene with 12.2 cM and 4.2 cM genetic distances, respectively (Figure 3B). Our linkage and genetic map analyses confirmed that *er1-*8 and *er1-*9 controlled *E. pisi* resistance in G0004389 and G0004400, respectively (Figure 3).

### 2.4. Development of Functional Markers for er1-8 and er1-9

The indel marker, InDel*-er1*-8 flanking the 3-bp deletion in *er1*-8, amplified 231-bp and 228-bp fragments in the parents WSU 28 and G0004389, respectively. The amplicons were clearly polymorphic between the contrasting parents, as visualized on an 8% polyacrylamide gel (Appendix A). InDel*-er1*-8 was then used to identify the genotypes of the 120 F_2_ plants derived from WSU 28 × G0004389. Three distinct electrophoretic bands corresponding to the homozygous resistant (R), homozygous susceptible (S), and heterozygous (H) genotypes were observed (Appendix A). Each F_2_ genotype corresponded to a phenotype of the 120 F_2:3_ families. A chi-squared (χ^2^) test showed that the segregation ratio of InDel*-er1*-8 in the F_2:3_ population derived from WSU 28 × G0004389 fit a 1:2:1 (χ^2^ = 0.48; *P* = 0.79). All results suggested that the marker InDel*-er1*-8 co-segregated with gene *er1*-8, indicating a co-dominant marker.

In the Kompetitive allele-specific PCR (KASPar) assay, KASPar-*er1-8* and KASPar-*er1-*9 successfully distinguished the contrasting parents (WSU 28 and G0004389, Bawan 6 and G0004400) into two different clusters corresponding to the FAM-labeled and HEX-labeled groups, respectively (Appendix A). When KASPar-*er1*-8 and KASPar-*er1*-9 were used to analyze the 120 and 119 F_2_ progeny derived from WSU 28 × G0004389 and Bawan 6 × G0004400, the KASPar markers clearly separated the F_2_ progeny into three clusters corresponding to three genotypes: homozygous resistant, homozygous susceptible, and heterozygous (Appendix A). In the F_2_ population derived from WSU 28 × G0004389, 30 plants were identified as homozygous resistant, 63 were heterozygous, and 27 were homozygous susceptible. In the F_2_ population derived from Bawan 6 × G0004400, 32 plants were homozygous resistant, 64 were heterozygous, and 23 were homozygous susceptible. These results were completely consistent with the phenotypes of both F_2:3_ populations, suggesting that KASPar-*er1*-8 and KASPar-*er1*-9 co-segregated with *er1*-8 and *er1*-9, respectively. A chi-squared (χ^2^) test showed that both segregation ratios of KASPar-*er1*-8 and KASPar-*er1*-9 in respective F_2_ populations fit 1:2:1 (KASPar-*er1*-8: χ^2^ = 0.48, *P* = 0.79; KASPar-*er1*-9: χ^2^ = 2.51; *P* = 0.29), indicating co-dominant markers.

### 2.5. Validation and Application of Functional Markers

Of the 169 germplasm accessions selected and tested for their phenotypic resistance to *E. pisi* isolate EPYN (Appendix A), 19 were phenotypically immune to *E. pisi*, 22 were resistant, and 128 were susceptible (Appendix A).

Among the 169 germplasms genotyped with InDel*-er1*-8, the 228-bp fragment corresponding to *er1-*8 was only amplified in G0004839 (Appendix A). In all of the other tested germplasm accessions, a 231-bp fragment was consistently amplified by InDel*-er1*-8, indicating that no accessions besides G0004839 carried *er1*-8 (Appendix A; Appendix A).

When the 169 germplasm accessions were genotyped with KASPar-*er1*-8, two distinct clusters were recovered, with one gene (*er1*-8) corresponding to G0004389 and the other (non-*er1*-8) to the other germplasms, respectively. Similarly, when the germplasms were genotyped with KASPar-*er1*-9, two distinct clusters were recovered, corresponding to G0004400 and all of the other germplasms, respectively (Appendix A; Appendix A). Thus, markers KASPar-*er1*-8 and KASPar-*er1*-9 effectively identified pea germplasms carrying the *er1*-8 and *er1*-9 alleles, respectively. Our results also showed that none of the other 169 pea germplasm accessions carried the *er1*-8 or *er1*-9 alleles.

## 3. Discussion

Powdery mildew induced by *E. pisi* DC. is a major disease on pea and causes considerable yield losses worldwide. The resistance gene *er1* is the most widely deployed gene controlling powdery mildew in pea cultivars worldwide. Furthermore, *er1* allelic diversity has been widely reported in pea [14,21,25,26,27,38,40,41,42,43,44,51].

To date, more than 40 *MLO* mutant alleles have been described in the monocotyledonous plant barley [52]. It is predicted that additional *er1* alleles resulting from natural mutations would be present among pea germplasms from around the world. As expected, we not only encountered the four known *er1* alleles (*er1*-1, *er1*-2, *er1*-6, and *er1*-7) across the 53 *E. pisi*-resistant pea germplasms*,* but we also discovered two novel *er1* alleles: *er1*-8 in germplasm G0004389 from Afghanistan and *er1-*9 in germplasm G0004400 from Australia (Table 1).

Among the nine known *er1* alleles, *er1*-1 and *er1*-2 are most commonly used in pea breeding programs because they confer stable resistance to *E. pisi* [14,25,38,51]. Our results indicated that these two alleles were common in the tested pea germplasm accessions resistance to *E. pisi*. The *er1*-1 allele was found in seven accessions (12.73%), and *er1*-2 was found in 37 accessions (67.27%) (Table 2). Among the 86 *E. pisi-*resistant pea accessions, *er1*-1 and *er1*-2 were identified in 10 (11.62%) and 42 (48.84%) accessions, respectively (Table 1). Previously, *er1*-1 has been identified in four *E. pisi-*resistant pea cultivars (JI1559, Tara, and Cooper from Canada; and Yunwan 8 from China), while *er1*-2 has been identified in seven *E. pisi-*resistant pea cultivars (Stratagem, Franklin, Dorian, Nadir, X9002, Xucai 1, and G0005576) [14,24,25,27,38]. Here, more *E. pisi-*resistant germplasm accessions carrying the *er1*-1 and *er1*-2 alleles were identified.

At the genomic level, seven alleles (*er1*-1/*er1mut1*, *er1*-3, *er1*-4, *er1*-5, *er1*-6, *er1*-9, and *er1*-10/*er1mut2*) are the result of point mutations in the exons of wild-type *PsMLO1*. Four alleles result from single base substitutions in wild-type *PsMLO1* cDNA: in *er1*-1, a C→G at position 680 (exon 6); in *er1*-5, a G→A at position 570 (exon 5); in *er1*-6, a T→C at position 1121 (exon 11); and in e*r1*-10, a G→A at position 939 (exon 10) (Appendix A) [14,27,38,40]. Three alleles result from single base deletions in wild-type *PsMLO1* cDNA, including ΔG at position 862 (exon 8) in *er1*-3; ΔA at position 91 (exon 1) in *er1*-4; and ΔT at position 928 (exon 10) in *er1*-9 identified in this study [14] Two alleles result from small fragment deletions in wild-type *PsMLO1* cDNA, including a 10-bp deletion of positions 111–120 (exon 1) in *er1*-7 [26]; and a 3-bp deletion of positions 1339–1341 (exon 15) in *er1*-8. To date, only the *er1*-11 mutation is known to have resulted from an intron mutation in *PsMLO1* (a 2-bp insertion in intron 14) [42,43], and only *er1*-2 results from a large indel of unknown size in wild-type *PsMLO1* cDNA [14,24,27].

Previous studies have indicated that the *er1*-2 allele produces three distinct *PsMLO1* transcripts [14,25,27,51]. Interestingly, this study observed that the *er1*-2 carried by the pea germplasm accession G0002860 produced four distinct *PsMLO1* transcripts. One of these transcripts was characterized by a 129-bp deletion, corresponding to the deletion of exon 13 (68 bp) and exon 14 (61 bp) from wild-type *PsMLO1* cDNA, indicating alternative splicing of exon skipping. Previously, two transcripts of *er1*-2 were observed to have large insertions (155-bp and 220-bp) based on comparisons with the transcripts of wild-type *PsMLO1* cDNA [14,24,25,27,51]. Here, we discovered that the 155-bp “insertion” in *er1*-2 resulted from a 192-bp insertion at position 1263 and a 37-bp deletion of positions 1263–1299 in exon 14 of wild-type *PsMLO1*, while the 220-bp “insertion” resulted from a 257-bp insertion at position 1263 and a 37-bp deletion of positions 1263–1299 in exon 14 of wild-type *PsMLO1*. Another alternative transcript of *er1*-2, an 87-bp “insertion”, was observed and resulted from a 192-bp insertion and a 37-bp deletion in exon 14 and a 68-bp deletion corresponding to exon 13 of wild-type *PsMLO1*. Our blast analysis indicated that the 192- and 257-bp insertions had 95% sequence identity with a five-part repetition in the pea genomic BAC sequence (GenBank accession number CU655882). These insertions were also highly similar (~85–87% identity) to a portion of the giant *Ogre* retrotransposons in the pea genome (GenBank accession numbers AY299395, AY299398, AY299397, and AY299394).

Based on 10 cloned sequences, several pea germplasms had frame shift mutations with small fragment indels (4-bp, 5-bp, or 16-bp) in one or two cloned *PsMLO1* cDNA sequences. Previously, a 5-bp (GTTAG) insertion was identified in G0001763 and G0003831; 11-bp (GTAGGAATAAG) and 13-bp (GTAATCTTATTAG) deletions were identified in G0003831 and G0001778; and a 16-bp (CTCATCTTCCTCCAGG) deletion was detected in G0001778 [27]. These small fragment indels in the *PsMLO1* cDNAs were assumed to have resulted from aberrant splicing events during transcription [27].

Alternative splicing in eukaryotes is a pervasive molecular mechanism that significantly increases transcriptome and proteome complexity [53]. Four main types of alternative splicing are known: exon skipping, alternative 5′ splice sites, alternative 3′ splice sites, and intron retention [54]. Exon skipping is common in humans, while intron retention is common in plants [55]. Alternative splicing is involved in many physiological processes, including response to biotic and abiotic stressors [56]. In the pea germplasms, three types of alternative splicing, intron retention, exon skipping, and alternative 5′ splice site selection, were observed in this study. Interestingly, pea germplasms carrying identical *er1* alleles varied in their resistance to *E. pisi*, from immune (disease severity of 0) to merely resistant (disease severity of 1–2) (Table 1). Alternative splicing in response to biotic stress may affect the expression of regulatory genes. Thus, it is speculated that the alternative splicing of *er1* alleles might affect the expression of the *E. pisi* resistance genes *er1*. In addition, the different levels of resistance to *E. pisi* might result from other related gene regulation. It is possible that multiple molecular processes and pathways contribute to *MLo*-based *E. pisi* resistance in peas.

Several functional markers specific to the previously recognized *er1* alleles have already been developed to facilitate marker-assisted breeding of pea cultivars resistant to *E. pisi* [14,24,26,27,40,42,43,44]. Pavan et al. [38] developed a functional cleaved amplified polymorphic sequence (CAPS) marker for *er1*-5, while Pavan et al. [44] developed functional markers for the five *er1* alleles, *er1*-1 through *er1*-5. Santo et al. [40] developed functional markers for *er1mut1* and *er1mut2,* and Wang et al. [24] developed a dominant marker for *er1*-2. Sudheesh et al. [43] developed a functional marker for *er1*-11, while Sun et al. [26,27] developed co-dominant functional markers for *er1*-6 and *er1*-7. More recently, Ma et al. [42] developed eight KASPar markers for eight known *er1* alleles, excluding *er1*-2.

This study discovered two novel *er1* alleles resulting from novel mutations of wild-type *PsMLO1* cDNA: *er1*-8 was generated by a 3-bp deletion in exon 15, and *er1-*9 was generated by a 1-bp deletion in exon 10. The co-dominant functional markers specific to *er1*-8 (InDel-*er1*-8 and KASPar-*er1*-8) and to *er1*-9 (KASPar-*er1*-9) were developed. These markers were validated in genetic populations and in pea germplasms. Our results are vital for future studies of powdery mildew resistance and for the development of *E. pisi*-resistant pea cultivars. The novel *er1* alleles and the corresponding co-dominant functional markers developed herein could constitute efficient and powerful tools for the breeding of *E. pisi*-resistant peas.

## 4. Materials and Methods

### 4.1. Plant Material and E. pisi Isolate

Previously, 86 pea germplasms had been found to be *E. pisi*-resistant in screenings of over 1000 pea accessions in a worldwide collection [27,48,50]. And, 31 of 86 resistant pea germplasms had been previously identified the *E. pisi-*resistant *er1* allele [24,25,26,27,48,51]. In this study, the remaining 55 of the 86 *E. pisi*-resistant pea germplasms from the United States of America, Canada, Germany, India, Australia, Columbia, England, Denmark, Nepal, Japan, Afghanistan, and Mexico, as well as data from the International Crop Research Institute for Semi-arid Tropics (ICRISAT) and conserved in the China National Genebank (http://www.cgris.net/), were used as research materials to reveal their *E. pisi-*resistant genes at *er1* locus (Table 1). The Chinese pea cultivars Bawan 6 and Longwan 1, which carry the *E. pisi-*susceptible gene *Er1,* were used as susceptible controls [24,51]. The Chinese pea cultivars Xucai 1, carrying *er1*-2, and YI (JI1591), carrying *er1*-4, were used as *E. pisi-*resistant controls [14,25].

The *E. pisi* isolate EPYN from Yunnan Province of China was used as the inoculum [26,27,41,48,50,51]. The EPYN isolate was maintained through continuous re-inoculation of seedlings of the pea cultivar Longwan 1 under controlled conditions. The inoculated plants were incubated in a growth chamber to prevent contamination with other isolates [25].

### 4.2. Phenotypic Evaluation

Twenty seeds were planted from each of the 55 *E. pisi-*resistant pea germplasm accessions, from the susceptible controls Bawan 6 and Longwan 1, and from the resistant controls Xucai 1 and YI [27]. The seedlings were thinned to 15 per pot before the phenotypic evaluation. Three replications were planted. Seeded pots were placed in a greenhouse maintained at 18 to 26 °C. At the same time, the *E. pisi* inoculum was prepared by inoculating the 10-day-old seedlings of the susceptible pea cultivar Longwan 1, which were incubated in a growth chamber at 20 ± 1 °C with a 12-h photoperiod. Two weeks later, the 14-day-old seedlings of 55 germplasm accessions and controls were inoculated by gently shaking off conidia of the Longwan 1 plants. Inoculated plants were incubated in a growth chamber at 20 ± 1 °C with a 12-h photoperiod. Ten days later, disease severity was rated based on a scale (0–4 scale) [27]. Plants with a score of 0 were considered *E. pisi-*immune, while those with scores of 1 and 2, 3 and 4 were considered as *E. pisi-*resistant and *E. pisi-*susceptible, respectively. For those identified as immune or resistant to *E. pisi,* repeated identification was performed.

### 4.3. RNA Extraction and PsMLO1 Sequence Analysis

The extraction of total RNA and synthesis of cDNA from the 55 pea germplasms and controls were completed according to our previous studies [25,26,27].

To identify the resistance alleles at the *er1* loci, the full-length cDNAs of the *PsMLO1* homologs were amplified using the primers specific for *PsMLO1* [14]. The PCR cycling conditions were as follows: 95 °C for 5 min; then 35 cycles of denaturation at 94 °C for 30 s, annealing at 58 °C for 45 s, and extension at 72 °C for 1 min; and a final extension at 72 °C for 10 min. The purified amplicons were cloned with a pEasy-T5 vector (TransGen Biotech, Beijing, China). The sequencing reactions of 10 clones per germplasm (including controls) were performed by the Shanghai Shenggong Biological Engineering Co., Ltd. (Shanghai, China). The resulting sequences were aligned with wild-type *PsMLO1* of pea (NCBI accession number: FJ463618.1) using DNAMAN v6.0 (Lynnon Biosoft, Quebec, Canada).

### 4.4. Genetic Analysis of Pea Germplasms Carrying Novel Alleles

To confirm the resistance genes, *er1*-8 and *er1*-9, G0004389 and G0004400 were crossed with the *E. pisi*-susceptible cultivars WSU 28 and Bawan 6, respectively, to generate genetic populations. The derived F_1_, F_2_, and F_2:3_ populations from both crosses (WSU 28 × G0004389 and Bawan 6 × G0004400) were used to evaluate the *E. pisi* resistance and genetic analysis of G0004389 and G0004400. The four parents and the derived F_1_ and F_2_ populations were planted in a propagation greenhouse to generate F_2_ and F_2:3_ family seeds, respectively.

Plants of the F2 populations at the fourth or fifth leaf stage were inoculated with the *E. pisi* isolate EPYN using the detached leaf method [25,26,27,57]. After inoculation, the treated leaves were placed in a growth chamber at 20 °C with a 14-h photoperiod. The four parents (WSU 28, G0004389, Bawan 6, and G0004400) were also inoculated as controls. Ten days after inoculation, disease severity was rated based on a scale of 0–4 as described above. Plants with scores of 0–2 and 3–4 were classified as resistant and susceptible, respectively [25,26,27,31,58]. Those plants identified as *E. pisi*-resistant were tested again to confirm their resistance.

Twenty-five seeds were selected randomly from each of the 120 F_2:3_ families derived from WSU 28 × G0004389, and from each of the 119 F_2:3_ families derived from Bawan 6 × G0004400. These seeds were planted and cultivated together with their parents, following previously published protocols [25,26,27]. Disease severity was scored 10 days after inoculation using the 0–4 scale, as described above for the phenotypic identification of the pea germplasms. The F_2:3_ families with scores of 0–2 and 3–4 were classified as homozygous resistant and homozygous susceptible, respectively. Families with scores of 0–2 and 3–4 were considered segregated to *E. pisi* resistance [27,31,58]. The families identified as homozygous resistant or resistance segregated were subjected to repeated testing.

A chi-squared (χ^2^) analysis was used to evaluate the goodness-of-fit to Mendelian segregation ratio of the F_2_ and F_2:3_ phenotypes derived from WSU 28 × G0004389 and Bawan 6 × G0004400.

### 4.5. Genetic Mapping of the Resistance Alleles er1-8 and er1-9

The Genomic DNA was isolated from the leaves of the F_2_ populations and of their parents using the cetyltrimethylammonium bromide (CTAB) extraction method [59]. The DNA solution was diluted and stored at −20 °C until use.

To map the novel *er1* alleles *er1*-8 and *er1*-9, the 10 known *er1*-linked markers on the pea LG VI, including four sequence-characterized amplified region (SCAR) markers [ScOPD10-650 [17], ScOPE16-1600 [18], ScOPO18-1200 [18], and ScOPX04-880 [23]; five simple sequence repeat (SSR) markers (PSMPSAD51, PSMPSA5, PSMPSAD60, i.e., AD60, PSMPSAA374e, and PSMPSAA369); a gene marker [Cytosine-5, DNA-methyltransferase (c5DNAmet)] [20,24,25,26,27,48,60]; and 10 additional molecular markers on the pea LG VI (AD160, AC74, AC10_1, AA224, AA200, AD159, AD59, AB71, AA335, and AB86), were used to screen for polymorphisms between the crossed parents (i.e., WSU 28 and G0004389; Bawan 6 and G0004400) [61]. The parental polymorphic markers were then used for genetic linkage analysis based on the genotype of each F_2_ plant. PCR amplification of each marker was conducted in a total volume of 20 µL according to the previous descriptions [25,26,27]. PCR reactions were performed in a thermal cycler (Biometra, Göttingen, Germany) [25,26,27]. The PCR products were separated on 6% polyacrylamide gels.

The segregation data of the polymorphic markers in the F_2_ populations were evaluated for goodness-of-fit to Mendelian segregation patterns with a chi-squared (χ^2^) test. Genetic linkage analyses were completed using MAPMAKER/EXP version 3.0b. A logarithm of odds (LOD) score > 3.0 and a distance < 50 cM were used as the thresholds to determine the linkage groups [62]. Genetic distances were determined using the Kosambi mapping function [63]. The genetic linkage map was constructed using the Microsoft Excel macro MapDraw [64].

### 4.6. Development of Functional Markers for er1-8 and er1-9

Primers flanking the mutation site (GTG/---) were designed based on the *PsMLO1* gene sequence (GenBank accession number KC466597), using Primer Premier v5.0, to develop an insertion/deletion (indel) functional marker specific to allele *er1*-8, InDel-*er1*-8 (Table 3). The marker InDel-*er1*-8 was used to determine the genotypes of the 120 F_2_ offspring derived from WSU 28 × G0004389. PCR amplification was performed as described above on a thermal cycler with the following cycling program: 95 °C for 5 min; 35 cycles of 94 °C for 30 s, 55 °C for 30 s, and 72 °C for 30 s; and 72 °C for 7 min. PCR products were separated on 8% polyacrylamide gels.

Based on allele *er1*-8 indels (a 3-bp deletion) and *er1*-9 SNPs (1-bp deletion) in *PsMLO1*, the forward primers and the common reverse primers specific to *er1*-8 (KASPar-*er1*-8) and *er1*-9 (KASPar-*er1*-9) were designed for Kompetitive allele-specific PCR (KASPar) markers by LGC KBioscience (KBioscience, Hoddesdon, UK), respectively. In brief, two KASPar markers (KASPar-*er1*-8 and KASPar-*er1*-9) were used to detect parental polymorphisms (WSU 28 × G0004389, and Bawan 6 × G0004400), and then used to analyze the genotypes of the F_2_ offspring (WSU 28 × G0004389: 120 F_2_ individuals; Bawan 6 × G0004400: 119 F_2_ individuals).

KASPar markers were amplified with a Douglas Scientific Array Tape Platform (China Golden Marker, Beijing, Biotech Co., Ltd.) in a 0.8 µL Array Tape reaction volume with 10 ng dry DNA, 0.8 µL 2 × KASP master mix, and 0.011 µL primer mix (KBioscience, Hoddesdon, UK). A Nexar Liquid handling instrument was used to add the PCR solution to the Array Tape (Douglas Scientific). PCRs were performed on a Soellex PCR Thermal Cycler with the following conditions: initial denaturation at 94 °C for 15 min; followed by 10 cycles of denaturation at 94 °C for 20 s, and 65 °C for 60 s at an annealing temperature that decreased by 0.8 °C per cycle; and then 26 cycles of denaturation at 94 °C for 20 s and 57 °C for 60 s; and a final cooling to 4 °C. A fluorescent end-point reading was completed with the Araya fluorescence detection system (part of the Douglas Scientific Array Tape Platform). Genotypes and clusters were visualized with Kraken (http://ccb.jhu.edu/software/kraken/MANUAL.html).

### 4.7. Validation and Application of Functional Markers

To test the efficacy of the novel functional markers specific to *er1*-8 (InDel-*er1*-8 and KASPar-*er1-8*) and *er1*-9 (KASPar-*er1*-9), 169 pea germplasm accessions were tested for (a) their phenotypic resistance to *E. pisi* isolate EPYN and (b) whether they carried the *er1* alleles *er1*-8 or *er1*-9 (Appendix A). The four parent cultivars (WSU 28, G0004389, Bawan 6, and G0004400) were used as contrasting controls, and seven cultivars, including Tara (*er1*-1) [41], Xucai 1 (*er1*-2) [25], JI210 (*er1*-3) [14], YI (*er1*-4) [14], G0001778 (*er1*-6) [27], DDR11 (*er1*-7) [26], and GI2480 (*er2*) [28], were used as positive controls (Appendix A).

DNA was extracted from the 169 selected pea germplasm accessions and the 11 controls (four parents and seven resistant cultivars with known *er1* alleles) using the CTAB method (Shure et al. 1983). PCR amplifications of the indel and KASPar markers were performed as described above (in the section “Development of functional *er1*-8 and *er1*-9 markers”).

## Figures and Tables

**Figure 1 ijms-20-05071-f001:**
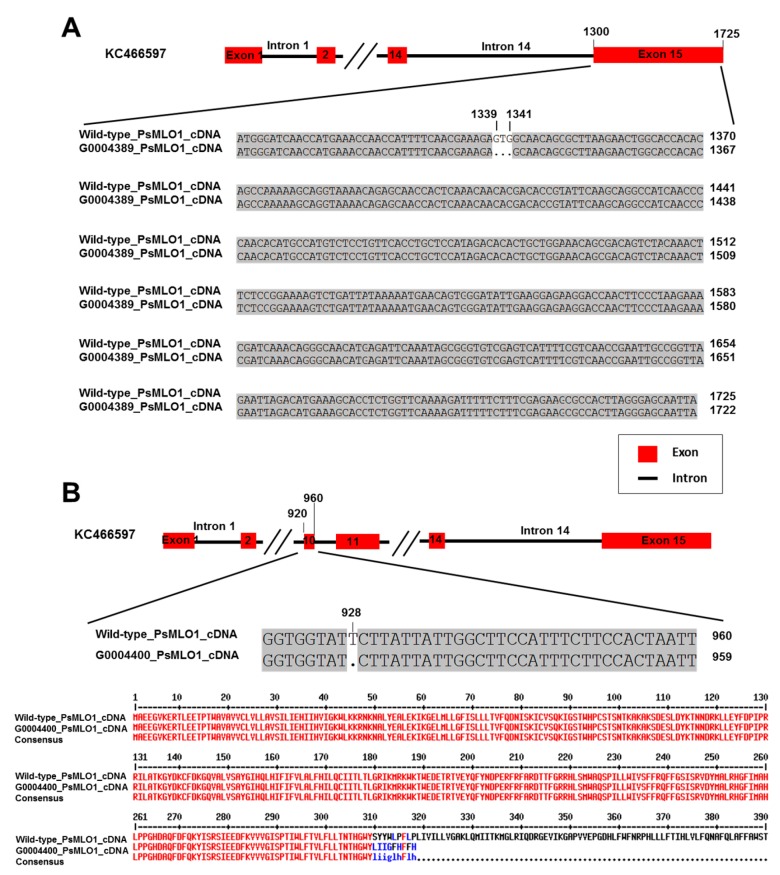
*PsMLO1* cDNA sequences from the powdery mildew-resistant pea germplasms G0004389 and the wild-type pea cultivar Sprinter (GenBank accession number: FJ463618.1), and *PsMLO1* cDNA sequences from G0004400 and amino acid sequence difference caused by mutation. (**A**) There is a 3-bp deletion (GTG) in the *PsMLO1* cDNA of G0004389 at positions 1339–1341 of exon 15. (**B**), there is a single base deletion (T) in the *PsMLO1* cDNA sequence of G0004400 at position 928 in exon 10, the lower figure shows the difference of amino acid sequence from G0004389 and the wild-type pea cultivar Sprinter. The two mutation sites are indicated in the respective cDNA sequences.

**Figure 2 ijms-20-05071-f002:**
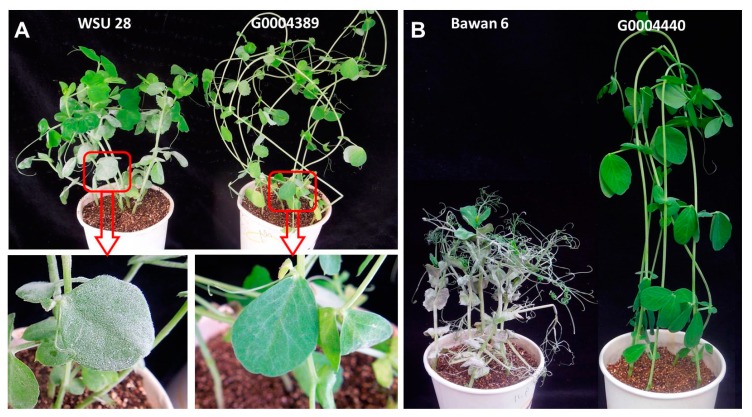
Phenotypic evaluation of the *Erysiphe pisi*-resistant pea germplasms G0004389 and G0004400, as well as the *E. pisi-*susceptible cultivars WSU 28 and Bawan 6, after inoculation with *E. pisi* isolate EPYN. (**A**) G0004389 and *E. pisi-*susceptible cultivar WSU 28. (**B**) G0004400 and *E. pisi-*susceptible cultivar Bawan 6.

**Figure 3 ijms-20-05071-f003:**
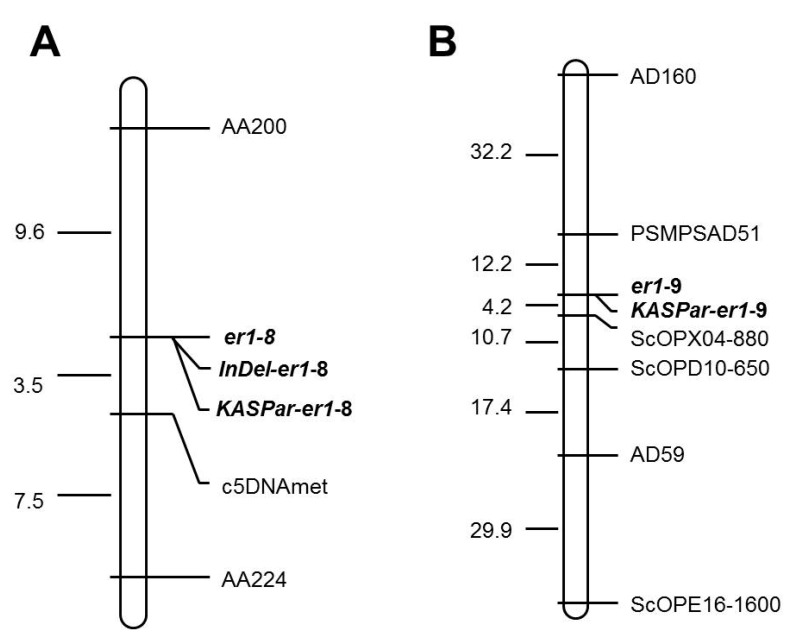
Genetic linkage maps constructed using the *er1*-linked markers and the functional markers for *er1*-8 and *er1*-9, based on the F_2_ populations derived from (**A**) WSU 28 × G0004389 and (**B**) Bawan 6 × G0004400. Map distances and loci order were determined with MAPMAKER v3.0 (Lander et al. 1993). Estimated genetic distances between loci are shown to the left of the maps in centiMorgans (cM).

**Table 1 ijms-20-05071-t001:** Information about phenotype and the resistance gene at the *er1* locus of the 86 *Erysiphe pisi*-resistant and the two *E. pisi*-susceptible controls (two controls are bolded).

No.	Accession No./Germplasm Name	Origin	Phenotype	*er1* Allele	Reference
1	G0004389	Afghanistan	I	*er1*-8	This study
2	G0004382	Australia	I	*er1*-1	This study
3	G0004400	Australia	I	*er1*-9	This study
4	G0004417	Australia	I	*er1*-2	This study
5	G0004434	Australia	I	*er1*-2	This study
6	G0004448	Australia	I	*er1*-2	This study
7	G0004450	Australia	I	*er1*-2	This study
8	G0002102	Canada	I	*er1*-6	This study
9	G0006514	Canada	R	*er1*-2	This study
10	G0006515	Canada	R	*er1*-2	This study
11	G0006516	Canada	I	*er1*-2	This study
12	G0006519	Canada	I	*er1*-2	This study
13	G0003925	Canada	I	*er1*-1	[41]
14	Cooper	Canada	I	*er1*-1	[41]
15	G0005576	China, Chongqing	I	*er1*-2	[27]
16	G0006273	China, Gansu	I	*er1*-2	[24]
17	20012	China, Gansu	I	*er1*-1	This study
18	Jia2	China, Gansu	I	*er1*-2	This study
19	Texuan11	China, Gansu	I	*er1*-2	This study
20	Hehuan66	China, Gansu	R	*er1*-1	This study
**21**	**Longwan 1**	**China, Gansu**	**S**	***Er1***	**[51]**
22	PI391630	China, Guangdong	I	*er1*-4	[14]
23	Xucai1	China, Hebei	I	*er1*-2	[25]
24	G0003694	China, Hebei	R	*er1*-6	[27]
**25**	**Bawan 6**	**China, Hebei**	**S**	***Er1***	**[24]**
26	L0314	China, Yunnan	I	*er1*-1	[51]
27	L1332	China, Yunnan	I	*er1*-2	[51]
28	L1335	China, Yunnan	I	*er1*-2	[51]
29	G0001747	China, Yunnan	R	*er1*-6	This study
30	G0001752	China, Yunnan	I	*er1*-6	[27]
31	G0001763	China, Yunnan	I	*er1*-6	[27]
32	G0001764	China, Yunnan	I	*er1*-6	[27]
33	G0001767	China, Yunnan	I	*er1*-6	[27]
34	G0001768	China, Yunnan	I	*er1*-6	[27]
35	G0001773	China, Yunnan	I	*er1*-6	This study
36	G0001777	China, Yunnan	I	*er1*-6	[27]
37	G0001778	China, Yunnan	I	*er1*-6	[27]
38	G0001780	China, Yunnan	I	*er1*-6	[27]
39	G0003824	China, Yunnan	R	*er1*-6	[27]
40	G0003825	China, Yunnan	I	*er1*-6	[27]
41	G0003826	China, Yunnan	I	*er1*-6	[27]
42	G0003831	China, Yunnan	R	*er1*-6	[27]
43	G0003834	China, Yunnan	R	*er1*-6	[27]
44	G0003836	China, Yunnan	R	*er1*-6	[27]
45	G0003839	China, Yunnan	R	*er1*-6	This study
46	G0005117	China, Yunnan	I	*er1*-6	This study
47	G0003974	China, Yunnan	I	*er1*-7	This study
48	G0003975	China, Yunnan	I	*er1*-7	This study
49	Yunwan4	China, Yunnan	R	*er1*-1	This study
50	Yunwan18	China, Yunnan	R	*er1*-2	This study
51	Yunwan35	China, Yunnan	I	*er1*-2	This study
52	Yunwan37	China, Yunnan	I	*er1*-6	This study
53	L2157	China, Yunnan	I	*er1*-2	This study
54	G0002848	Denmark	I	*er1*-2	This study
55	G0002971	England	I	*er1*-2	This study
56	G0002859	Germany	I	*er1*-2	This study
57	G0002860	Germany	I	*er1*-2	This study
58	G0002883	Germany	I	*er1*-2	This study
59	G0003895	ICRISAT	I	e*r1-*7	[26]
60	G0003897	ICRISAT	I	*er1*-2	This study
61	G0003899	ICRISAT	I	*er1*-7	[26]
62	G0003907	ICRISAT	I	*er1*-2	This study
63	G0003911	ICRISAT	I	*er1-*2	This study
64	G0003961	India	I	*er1*-2	This study
65	G0003967	India	I	*er1*-7	[26]
66	G0003958	India	I	*er1*-7	[26]
67	G0006285	Japan	R	*er1*-2	This study
68	G0004332	Mexico	R	*er1*-1	This study
69	G0004394	Nepal	R	*er1*-7	[26]
70	G0002980	Unknown country	I	*er1*-2	This study
71	G0003931	Unknown country	I	*er1*-7	[26]
72	G0003935	Unknown country	I	*er1*-2	This study
73	G0003936	Unknown country	I	*er1*-7	[26]
74	G0003942	Unknown country	I	*er1*-1	This study
75	G0003943	Unknown country	I	*er1*-1	This study
76	G0002128	USA	I	*er1*-2	This study
77	G0002129	USA	I	*er1*-2	This study
78	G0002131	USA	I	*er1*-2	This study
79	G0002132	USA	I	*er1*-2	This study
80	G0002134	USA	I	*er1*-2	This study
81	G0002137	USA	I	*er1*-2	This study
82	G0002183	USA	I	*er1*-2	This study
83	G0002235	USA	I	*er1*-6	This study
84	G0002250	USA	I	*er1*-2	This study
85	G0002602	USA	I	*er1*-2	This study
86	G0002608	USA	I	*er1*-2	This study
87	G0002847	USA	I	*er1*-2	This study
88	G0002960	USA	I	*er1*-2	This study

“R”, “I”, and “S” stand for resistant, immune, and susceptible, respectively.

**Table 2 ijms-20-05071-t002:** The distribution and numbers of pea germplasm accessions carrying *er1* alleles.

Country	No. of Pea Germplasm Accessions Contained *er1* Alleles									
	*er1*-1	*er1*-2	*er1*-3	*er1*-4	*er1*-5	*er1*-6	*er1*-7	*er1*-8	*er1*-9	Total
USA	-	12	-	-	-	1	-	-	-	13
Canada	-	4	-	-	-	1	-	-	-	5
Germany	-	3	-	-	-	-	-	-	-	3
ICRISAT	-	3	-	-	-	-	-	-	-	3
India	-	1	-	-	-	-	-	-	-	1
Australia	1	4	-	-	-	-	-	-	1	6
England	-	1	-	-	-	-	-	-	-	1
Denmark	-	1	-	-	-	-	-	-	-	1
Nepal	-	-	-	-	-	-	-	-	-	0
Japan	-	1	-	-	-	-	-	-	-	1
Afghanistan	-	-	-	-	-	-	-	1	-	1
Mexico	1	-	-	-	-	-	-	-	-	1
China	3	5	-	-	-	5	2	-	-	15
Unknown country	2	2	-	-	-	-	-	-	-	4
Total	7	37	-	-	-	7	2	1	1	55

“-” indicates there was no pea germplasm containing this *er1* allele.

**Table 3 ijms-20-05071-t003:** Sequence information for the indel and Kompetitive allele-specific PCR (KASPar) markers specific to *er1*-8, and for the KASPar marker specific to *er1*-9.

Markers	Primers	Sequence Information (5′-3′)	Annealing Tm
InDel-*er1-8*	Forward	GTTTTGACTGATATGACAGATGGGA	55 °C
	Reverse	GTTTGTAGACTGTCGCTGTTTCC	
KASPar-*er1-8*	Forward-TGG	TGGCAACAGCGCTTAAGAACTGG	65–57 °C touchdown
	Forward	GAGCAACAGCGCTTAAGAACTGG	
	Common reverse	TGGTTGGTTTCATGGTTGATCCCATC	
KASPar-*er1-9*	Forward-T	TTTTGTTATATGGGCAGGGTGGTATT	65–57 °C touchdown
	Forward	TGTTATATGGGCAGGGTGGTATC	
	Common reverse	CAAAATGTAGATTATGCTTACAATTAGTGGA

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
