# Peer review of "Two Novel er1 Alleles Conferring Powdery Mildew (Erysiphe pisi) Resistance Identified in a Worldwide Collection of Pea (Pisum sativum L.) Germplasms"

_ijms, 2019, doi:10.3390/ijms20205071_

Round 1

Reviewer 1 Report

The MS represent allele mining efforts performed in pea for the very well studied er1/PsMLO1 gene. 

Author Response

Thank you for your comments.

Reviewer 2 Report

The paper reports the identification of two new er1 alleles conferring resistance to powdery mildew in Pisum sativum. They have proven the resistance is due to mutations at the MLO1/er1 locus by analyzing resistant and susceptible segregants from a cross. The authors have also classified the MLO1/er1 allele constitution of 86 E. pisi-resistant lines. The er1 mutations in the two new resistance alleles were identified to be a 3-bp deletion and a 1 bp deletion. The 1 bp. deletion changes the reading frame of the PsMLO1 coding sequence and probably leads to a truncated translation product (but this is not shown). My only criticisms of the paper are in the way the data is presented; In Figure 1B is necessary to show the effect of the 1 bp deletion at the level of the protein sequence.

In addition, it would be helpful to have all of the mutations listed in the paper mapped to a figure showing the gene structure.; at the moment they are just treated in the text.

Author Response

Revierwer 2

The paper reports the identification of two new er1 alleles conferring resistance to powdery mildew in Pisum sativum. They have proven the resistance is due to mutations at the MLO1/er1 locus by analyzing resistant and susceptible segregants from a cross. The authors have also classified the MLO1/er1 allele constitution of 86 E. pisi-resistant lines. The er1 mutations in the two new resistance alleles were identified to be a 3-bp deletion and a 1 bp deletion. The 1 bp. deletion changes the reading frame of the PsMLO1 coding sequence and probably leads to a truncated translation product (but this is not shown). My only criticisms of the paper are in the way the data is presented;

Comment 1: In Figure 1B is necessary to show the effect of the 1 bp deletion at the level of the protein sequence.

Response: As you suggested, we add information about amino acid sequence difference caused by 1 bp deletion, to show the effect of the 1 bp deletion at the level of the protein sequence.

Comment 2: In addition, it would be helpful to have all of the mutations listed in the paper mapped to a figure showing the gene structure.; at the moment they are just treated in the text.

Response: As you suggested, we added a figure (Figure S1) to show the mutation positions of all er1 alleles, and all er1 alleles were mapped to the er1 gene structure.  Thanks for your professional suggestion and good comments.